# Biopolymer Chitosan Surface Engineering with Magnesium Oxide-Pluronic-F127-Escin Nanoparticles on Human Breast Carcinoma Cell Line and Microbial Strains

**DOI:** 10.3390/nano13071227

**Published:** 2023-03-30

**Authors:** Suresh Mickymaray, Mohammed Saleh Al Aboody, Mostafa M. Eraqi, Wardah. A. Alhoqail, Abdulaziz S. Alothaim, Kaviya Suresh

**Affiliations:** 1Department of Biology, College of Science, Al Zulfi, Majmaah University, Majmaah 11952, Saudi Arabia; 2Centre of Molecular Medicine and Diagnostics (COMManD), Saveetha Dental College and Hospitals, Saveetha Institute of Medical and Technical Sciences, Saveetha University, Chennai 602105, India; 3Microbiology and Immunology Department, Veterinary Research Institute, National Research Centre, Dokki, Giza 12622, Egypt; 4Department of Biology, College of Education, Majmaah University, Majmaah 11952, Saudi Arabia; 5Department of Pharmaceutics, Sri Ramachandra Faculty of Pharmacy, Sri Ramachandra Institute of Higher Education and Research (DU), Chennai 600116, India

**Keywords:** Pluronic F-127, magnesium oxide nanoparticles, escin, chitosan, MDA-MB-231 cancer cells, antimicrobial

## Abstract

Nanotechnology has been recognized as a highly interdisciplinary field of the twenty-first century, with diverse applications in biotechnology, healthcare, and material science. One of the most commonly employed non-toxic nanoparticles, magnesium oxide nanoparticles (MgO NPs), is simple, inexpensive, biocompatible, and biodegradable. Several researchers are interested in the biosynthesis process of MgO NPs through chemical and physical approaches. This is because of their simplicity, affordability, and environmental safety. In the current study, green MgO-Chitosan-Pluronic F127-Escin (MCsPFE) NPs have been synthesized and characterized via various techniques like UV-visible, Fourier-transform infrared spectroscopy, Energy dispersive X-ray composition analysis, Transmission electron microscopy, field emission scanning electron microscopy, X-ray Diffraction, Photoluminescence, and Dynamic light scattering analyses. The average crystallite size of MCsPFE NPs was 46 nm, and a face-centered cubic crystalline structure was observed. Further, the antimicrobial effectiveness of NPs against diverse pathogens has been assessed. The cytotoxic potential of the nanoparticles against MDA-MB-231 cell lines was evaluated using the MTT test, dual AO/EB, JC-1, DCFH–DA, and DAPI staining procedures. High antimicrobial efficacy of MCsPFE NPs against Gram-positive and Gram-negative bacterial strains as well as *Candida albicans* was observed. The findings concluded that the NPs augmented the ROS levels in the cells and altered the Δψm, leading to the initiation of the intrinsic apoptotic cell death pathway. Thus, green MCsPFE NPs possess immense potential to be employed as an effective antimicrobial and anticancer treatment option.

## 1. Introduction

An age of innovation in illness prevention and treatment has been introduced by nanotechnology. Nanoparticles (NPs), which are regarded as nanometer-sized particles ranging in the dimensions of 1–100 nm, have been successfully applied in the specialized fields of cancer therapy, antimicrobial treatment, biosensor-based therapy, etc. A nanoparticle form has an increased capacity for resisting oxidation and deactivation as well as time-delayed absorption across various membrane barriers [1]. Biotechnology, healthcare, energy, and material science are just a few domains where nanotechnology has achieved widespread applicability [2].

Both traditional chemical and physical processes produce NPs, but they are difficult to scale up and require energy and toxic chemicals for capping. Furthermore, these procedures endanger nanoparticle biocompatibility, which is crucial [3] because hazardous substances remain on the surface of NPs even after repeated washings. Consequently, scientists focus on synthesizing nanomaterials from biogenic sources, including bacteria, plants, and algae. Plant-based biological approaches provide benefits, including one-pot synthesis, robustness, affordability, and environmental friendliness. In addition, they utilize the plant’s biological molecules as a capping compound and reducing agent. These macromolecules are safe and ecologically beneficial [4,5].

Metal oxide NPs have recently attracted considerable interest in the domains of biosensing, diagnostic equipment, catalysts, antitumor, and antimicrobial agents owing to their unique physical and chemical characteristics [6]. Magnesium oxide (MgO) NPs, a metal oxide nanoparticle, have been employed to treat various diseases due to their biocompatibility, decent functionality, low density, non-toxicity, recycling activity, biodegradability, and remarkable endurance in severe environments [7,8]. In addition, these NPs possess notable medical uses, including treating heartburn and bone regeneration and use as antitumor and antibacterial agents [9]. Moreover, these NPs are easy and inexpensive to synthesize and possess antibacterial potential [10].

The second most prevalent biopolymer on earth is Chitosan. It exhibits antibacterial properties and is safe, biocompatible, and biodegradable. It has a variety of applications that have been well studied, including as a biosorbent, chelating agent, and scaffolding for other nanomaterials. Natural polymers like Chitosan and phytocompounds like Escin have anti-oxidant effects; hence it has been attempted to integrate these substances into MgO nanoparticles in this study [11,12]. Additionally, for effective drug administration and delivery, Pluronic F-127 (PF-127), a triblock amphiphilic copolymer, has been recognized as an excellent medium because of its solubilizing capacity, biocompatibility, reverse gelation, and low toxicity [13].

*Aesculus hippocastanum* L., belonging to the Hippocastanaceae family and commonly referred to as horse chestnut, is a native of Western Asia. In addition to treating leg ulcers and varicose veins, hemorrhoids, cellulitis, hematomas, edema, venous insufficiency, pain, and swelling, it may also reduce swelling and pain [14]. Escin, a combination of triterpene saponins, is the primary active component of *A. hippocastanum*. It exhibits a range of pharmacological effects, including venotonic, antisecretory, anti-edematous, anti-inflammatory, and anti-oxidative effects, which are most commonly observed in seed cotyledons, although small amounts are also found in the leaves, bark, and pericarp of immature fruit [15]. In addition, Escin has been incorporated into the synthesis of several nanoparticles, including ZnO, CuO, and Au NPs [16,17]. However, it has not yet been used for MgO NP biosynthesis. 

In the past, several plants, including *Limonia acidissima*, *Costus pictus*, *Nephelium lappaceum* peels, *Artemisia abrotanum*, *Andrographis paniculata*, *Pisonia alba*, *Ocimum americanum*, orange fruit waste, gum acacia, and neem leaves have been employed in a few procedures for MgO NPs green synthesis and are reported earlier in the literature [18,19,20]. However, the *Aesculus hippocastanum*-derived compound, Escin, has not yet been utilized for MgO NP biosynthesis. Hence, the biosynthesis of MgO-Chitosan-Pluronic F127-Escin NPs and their biomedical application has been studied in the present investigation. Further, their antimicrobial activity against selected pathogens as well as anticancer potential against MDA-MB-231 (human breast cancer cells) cell lines, has been evaluated.

## 2. Materials and Methods

### 2.1. Materials

Chemicals including Magnesium (II) nitrate hexahydrate (Mg(NO_3_)_2_·6H_2_O), Chitosan, Pluronic F-127, and Escin were procured from Sigma-Aldrich, St. Louis, MO, USA. Sigma-Aldrich, St. Louis, MO, USA., provided culture consumables and media. Abcam, Cambridge, UK, provided the ELISA kits.

### 2.2. Preparation of MgO-Chitosan-Pluronic F127-Escin NPs

For preparing MgO-Chitosan-Pluronic F127-Escin (MCsPFE) NPs, 0.1M of magnesium (II) nitrate hexahydrate (Mg(NO_3_)_2_·6H_2_O), 0.5 g of Pluronic F-127, and 500 mg of Chitosan were suspended in an aqueous solution (50 mL) with 1% acetic acid. Further, the phytocomponent Escin (50 mg) was combined with the Mg-Chitosan-PF-127 solution. Later, a NaOH solution (0.1 M) was added to the MgO-Chitosan-PF-127-Escin solution. The resultant white solution was agitated continually at 80 °C for 5 h. The white precipitate was allowed to dry at 120° C for 1 h. The formed MCsPFE NPs in powder form were annealed at 200 °C for 5 h and collected for subsequent usage.

### 2.3. Characterization of MgO-Chitosan-Pluronic F127-Escin NPs

The MCsPFE NPs were characterized with an X-ray diffractometer (XRD) (Bruker-model: AXS D5005, Karlsruhe, Germany). The analysis was accomplished using Cu-Kα radiation with a wavelength of λ = 0.1541 nm, and it was scanned at an angle of 2θ between 20 and 80°. A dynamic light scattering (DLS) study was carried out to assess the average particle size of the NPs. The examination was performed using nanoparticles suspended in de-ionized distilled water, and the study was conducted maintaining a 90° scattering angle and at 25 °C. The size and surface morphologies of MCsPFE NPs were detected with FESEM investigation utilizing a Hitachi s-4800II (Hitachi, Tokyo, Japan) equipped with EDAX, which was employed to examine the elemental composition of the formulated NPs. At a potential of 20 KV, a photomicrograph was captured. Transmission electron microscopy (TEM) (Jeol Jem-2010F-Jeol, Akishima, Tokyo, Japan) was used to investigate the NP morphology in greater detail. In brief, the NPs were smeared on a copper grid, and electronic radiation was used to light it under a vacuum. Further, an electron beam was used to capture microphotographs after passing through the material. Utilizing Fourier-transform infrared spectroscopy (FTIR) spectroscopy (NicoletiS50-Thermo Fisher Scientific Corporation, Waltham, MA, USA) the functional groups present in the MCsPFE NPs were examined. The IR spectra were identified using the reflectance approach. The produced NPs were ground with potassium bromide (KBr) at a 1:100 ratio, and the blend was flattened into a disc to make KBr discs. The discs were directly introduced into the spectrometer and scanned in the 400–4000 cm^–1^ wave number range. The produced NPs were subjected to double-beam UV-visible spectroscopy (Shimadzu UV-2550, Scinteck Instruments, Virginia, Washington, DC, USA) analysis to confirm NP formation and identify the surface plasmon resonance peak. The nanomaterials were examined in the 1200–200 nm wavelength range, and experimentation was conducted thrice to establish the mean absorbance of the NPs.

The optical properties of the NPs were studied by photoluminescence spectroscopy (Roithner Lasertechnik, Vienna, Austria). The spectra were examined using the 350–550 nm spectral area, acquired at λexc = 470 nm. Three runs of the experiment were completed, and the energy band gap was computed by calculating the Tauc plot with the equation below:(αhν)^1/2^ = A (hν-E_g_)(1)
where h—Planck’s constant; A—constant proportion; α—absorption coefficient; v—vibration frequency; E_g_—band gap; n—sample transition nature. 

### 2.4. Evaluation of Antimicrobial Efficacy

The antibacterial effectiveness of the MCsPFE NPs against *Escherichia coli* (ATCC: 25922); *Enterobacter cloacae* (Clinical isolate); *Pseudomonas aeruginosa* (ATCC: 27853); *Staphylococcus aureus* (ATCC: 25923) and yeast fungi *Candida albicans* (ATCC: 10231) was assessed by the disc diffusion procedure. For this study, molten Muller–Hinton Agar was employed (Hi-Media, Mumbai, India). Employing micropipettes, test samples at 1, 1.5, and 2 mg/mL concentrations were distributed onto 6 mm diameter paper discs. They were placed over culture-seeded plates with control discs. Pretested disc potency passed Ciprofloxacin (5 µg) and Fluconazole (25 µg) antimicrobial discs (Hi-Media, Mumbai, India) were used as a positive control against bacterial and fungal pathogens, respectively. After that, the plates were incubated at 37 °C for 24 h. The inhibition zone diameter was recorded after the incubation period. The assay was performed in triplicates, and the mean inhibitory values were recorded in diameters (mm) with standard deviation. A confidence level of 95% (5% statistical significance) is used for data representation.

### 2.5. In Vitro Anticancer Activity of MgO-Chitosan-Pluronic F127-Escin NPs 

#### 2.5.1. Maintenance of MDA-MB-231 Cell Lines 

MDA-MB-231 (human breast cancer cell line) cells were kept in Dulbecco’s modified Eagle’s medium with additives such as 2 mM L-glutamine, 1.5 g/L Na_2_CO_3_, 0.1 mM nonessential amino acids, 2 mM L-glutamine, 1 mM sodium pyruvate, 1.5 g/L glucose, 10 mM HEPES buffer, and 10% fetal bovine serum.

#### 2.5.2. Cytotoxicity Assay

A cell viability assessment was conducted to examine MCsPFE NP anticancer potential. In a 24-well culture plate, MDA-MB-231 cells were introduced before being cultured for 12 h at room temperature with 5% CO_2_. Later, the cells were treated with the following concentrations of the NPs—1.56, 3.125, 6.25, 12.5, 25, 50, 100, and 200 µg/mL, and incubated at 37 °C for 24, 48, and 72 h. Later, the cells were subjected to 3-(4,5-dimethylthiazoyl-2-yl)-2,5-diphenyltetrazolium bromide (MTT) treatment (5 mg/mL) the following day, and they were further incubated for 3 h. Later, Dimethyl sulfoxide was utilized to suspend the formazan crystals developed by the MTT reaction before the optical density reading at 540 nm was examined with a microplate reader. Using the program Originpro8, the IC50 value of the NPs was determined. Three triple runs of the experiments were completed.

#### 2.5.3. Analysis of Apoptotic Cells by Dual Staining Approach

The dual staining approach was employed to assess the apoptotic efficiency of MCsPFE NPs against MDA-MB-231 cell lines. After being plated into a 6-well growth plate, the cells were exposed to 10 and 20 μg/mL of MCsPFE NPs for 24 h. After NP treatment, the cells were treated with an equal ratio of acridine orange/ethidium bromide (AO/EB) solution and subjected to incubation for 5 min. The cells were washed in PBS. After rinsing the cells in PBS, they were examined under a fluorescence microscope.

#### 2.5.4. Evaluation of Mitochondrial Membrane Potential (Δψm) by JC-1 (5,5′,6,6′-TetraChloro-1,1′,3,3′-Tetraethyl-Benzimidazolyl Carbocyanine Iodide) Staining

MDA-MB-231 cells (2 × 10^5^ cells/well) were plated onto 6-well plates and exposed to MCsPFE NPs (10 and 20 μg/mL) for 24 h. The cells were collected and stained with JC-1 (12.5 μM) dye in the medium for 20 min. After incubation, the mitochondrial labeled culture media was removed, cells were rinsed with PBS, fresh medium was added, and photographs were taken with an inverted fluorescence microscope.

#### 2.5.5. Detection of Reactive Oxygen Species (ROS) by DCFH–DA (Dichloro-Dihydro-Fluorescein Diacetate) Staining

Inoculated MDA-MB-231 cells (2 × 10^5^ cells/mL) were plated with new media and treated with MCsPFE NPs (10 and 20 µg/mL). After 24 h, the cells were harvested and subjected to three-time 1X PBS washes; the cells were treated with DCFH–DA (10 µmol/L) and left for incubation at room temperature for 20 min. To completely remove DCFH-DA, the cells were rinsed three times in cell culture media. Cellular ROS levels were then measured by fluorescence correlation microscopy.

#### 2.5.6. Analysis of Cell Morphology by DAPI (4′,6-Diamidino-2-Phenylindole) Staining

With DAPI staining, cell nuclear morphology was assessed using fluorescence microscopy. MCsPFE NPs (10 and 20 µg/mL) were introduced to the cancer cells and subjected to incubation for 24 h. The cells were then fixed with ice-cold ethanol (70%) and reconstituted in DAPI before being rinsed with PBS (pH 7.4) and kept for 15 min while covered in aluminum foil. The cells were later analyzed using a Nikon Eclipse Ti fluorescent microscope after being rinsed with PBS.

### 2.6. Statistical Analysis 

A post hoc Tukey post-test and a one-way ANOVA were used in the statistical analysis of all studies conducted in triplicate. Results were presented as mean ± SD and statistical significance was determined at *p* < 0.05.

## 3. Results and Discussion

### 3.1. Characterization of Green MgO-Chitosan-Pluronic F127-Escin NPs

#### 3.1.1. UV-Visible Spectroscopy

Using a UV-Vis spectrometer, the optical characteristics of synthesized MCsPFE NPs were investigated. UV-visible wavelengths were recorded from 200–1100 nm. The spectrum of UV-visible absorption capability of MCsPFE NPs in the UV and visible range is depicted in Figure 1a. The figure demonstrates that the synthesis of MCsPFE NPs was validated by UV-Vis spectra, which displayed broad absorption peaks at 254 nm. To investigate NP stability and its impacts on particle size distribution, key synthesis-related variables, such as temperature, heating duration, and precursor concentration, were investigated [21].

#### 3.1.2. FTIR Spectroscopic Analysis

As depicted in Figure 1b, FTIR spectroscopy is an appropriate method for identifying MCsPFE NP surface functionality. The Chitosan molecule overlaps with the broad intermolecular OH and -NH stretching bands, observed at 3435 cm^−1^. The carbonyl group of the amide vibration-NH group is located at 1646 cm^−1^. For Chitosan-coated with MgO, the CH2 bending vibration was maintained at 1468 cm^−1^. With respect to the functional groups in this polymer, the signals from the Pluronic F-127 and Escin molecules occurred at 2924 cm^−1^ for asymmetric stretching and 2854 cm^−1^ for symmetric stretching [22]. The presence of physisorbed and chemisorbed H-O-H causes a large envelope between them. The peak at 871 cm^−1^ suggests that the MgO NPs were synthesized completely. Metal and oxygen excitation peaks were detected at 766, 696, and 431 cm^−1^.

#### 3.1.3. PL Spectrum

The PL emission spectra of the MCsPFE NPs are illustrated in Figure 1c. The excitation wavelength of the MCsPFE NPs was 325 nm. Peaks of PL emission from MCsPFE NPs were found at 364, 380, 399, 412, 421, 429, 438, 453, and 482 nm. At 364, 380, and 399 nm, UV emission (NBE emission) could be observed. At 421, 429, 438, and 453 nm, the violet emission peak was detected, revealing the band-to-band transition. The strong recombination of electrons with oxygen vacancies, which might be produced, explains the blue-green emission observed at 482 nm [23,24,25].

### 3.2. X-ray Diffraction Examination 

Figure 2a displays the XRD patterns of the synthesized MCsPFE NPs. The distinctive diffraction peak for Chitosan can be seen in the XRD data at 19.87° [26]. The MgO diffraction peaks for the MCsPFE occurred at two values of 36.77, 42.57, 62.13, 74.55, and 78.42°, which can be ascribed to the (111), (200), (220), (311), and (222) (hkl) crystal planes, respectively, of the Face Centered Cubic (FCC) structure. The FCC structure of the MgO NPs was determined to have the highest intensity peak at the (200) (hkl) plane (JCPDS No. 45-0946) [26], which is similar to the crystal structure of the previously reported MgO-Chitosan-Pluronic F127-Escin. The whole diffraction peak depicted in the figure demonstrated the highly crystalline phase of the MgO-Chitosan-Pluronic F127-Escin NPs. The synthesized MCsPFE NPs were crystalline and devoid of impurities because no further peaks were visible in the spectrum for any other phases. Using the Scherrer equation, the average crystallite size of the MCsPFE NPs was analyzed as follows [27]:(2)$$\mathrm{Average}\;\mathrm{crystallite}\;\mathrm{size}\;D=\frac{k\lambda }{\beta_{Dcos\theta }}$$
where *λ* is the X-ray wavelength (1.54060 Å), *θ* is Bragg’s diffraction angle, and *β* is the angular peak width at half maximum (radians). The average particle size of green MCsPFE NPs is 46 nm. After successful synthesis, Debye–Scherrer’s equation and an examination of the material’s crystallinity were used to assess the average particle size of the synthesized NPs. Scherrer’s equation is considered the most crucial and frequently used equation to quantify particle size since it incorporates the 2θ and full width at half maximum values from the XRD data. 

### 3.3. Dynamic Light Scattering (DLS) Analysis

As can be observed in Figure 2b, the hydrodynamic size of MCsPFE NPs was calculated through DLS to determine particle size. The DLS particle size was larger than that of the XRD results since a water medium encircled the MCsPFE NPs, measured at 101 nm in size. This size is termed hydrodynamic size.

### 3.4. Morphology and Chemical Composition

The morphology of the green MCsPFE NPs surface was examined using the FESEM/TEM/SEAD pattern at various magnifications, ranging from 200 to 5 nm (Figure 3a–f). The polymers (Chitosan and Pluronic F127) were coated with MgO NPs, and the produced NPs appear rod-like in the FESEM and TEM image displays. Particle sizes range from 20–50 nm on average. Analyzing the surfaces of passivated MgO nanoparticles coated with Chitosan, Pluronic F-127, and Escin revealed significant surface roughness in aggregated formations. An EDAX spectrum was utilized to determine the chemical constitution of the produced MCsPFE NPs, as depicted in Figure 3f. The MCsPFE NPs have the following atomic percentages: 11.62% (C), 19.29% (N), 22.46% (Mg), and 46.62% (O). These carbon and nitrogen molecules originated from the MCsPFE NPs matrix components Chitosan, Pluronic F-127, and Escin.

### 3.5. Antimicrobial Activity of MCsPFE NPs

A disc diffusion approach was employed to assess the antimicrobial properties of MCsPFE NPs against the selected Gram-positive and Gram-negative bacterial strains and the yeast fungi *Candida albicans* at concentrations ranging from 1, 1.5, and 2 mg/mL (Figure 4 and Table 1). Figure 4 displays the zones of inhibition (ZOI) for MCsPFE NPs and the standard control discs Ciprofloxacin (5 µg/mL) and Fluconazole (25 µg/mL). The standard control antimicrobial agents and MCsPFE NPs both exhibited significant antimicrobial properties. It was noticed that the antimicrobial action improved as the NPs concentration increased. A number of processes, such as decreased particle size, the production of positively charged metal ions, the formation of ROS, etc., have been postulated in earlier studies to explain the antibacterial and antifungal actions of MgO-based NPs. Additionally, it was claimed that the small particle size of MCsPFE NPs contributes immensely to their antibacterial action (45 nm). The MCsPFE NPs lead to the cell death of microorganisms by destroying cell constituents, including DNA, proteins, and lipids, since they penetrate the cell membrane [28]. A similar concentration-based antibacterial effect has been observed by MnO nanoparticles against several pathogenic bacteria and has been reported [29]. Electrostatic interaction between MnO NAPs and negatively charged bacterial cell membranes causes them to come into contact [26]. They then attach to the cell membrane, which causes it to shift from being well-ordered to being disorderly. The cell membrane becomes less permeable as a result of this modification, allowing bacterial cell electrolytes to leak out and further destroying the mesosomes’ structure and ability to function [29]. Moreover, Chitosan serves as a catalyst in the cell’s destruction process. The positively charged Chitosan molecules associate with the negatively charged bacterial cell membrane, resulting in the leaking of cellular constituents [26].

### 3.6. Anticancer Activity of MgO-Chitosan-Pluronic F127-Escin NPs by Cytotoxicity Assay

Breast cancer is the second leading cause of women’s mortality [30]. Despite the fact that several cytotoxic medications are released on the market every year, the breast cancer burden is rising. This is because these treatments have unpredictable adverse effects or limited effectiveness. As a result, there is a growing need for therapeutic approaches that are both effective and have fewer adverse effects when treating malignancies [31]. In this aspect, polymer-based drug delivery systems can deliver precisely focused drug concentrations to tumorous areas without harming healthy cells [32]. Cytotoxicity assessment demonstrated a substantial reduction in breast cancer cell proliferation. The NP treatments exhibited concentration-dependent effects on the cytotoxicity, which is analogous to findings of antibacterial activity (Figure 5). A similar dose-dependent cytotoxic effect against breast cancer cells and skin melanoma cells has been reported by green MgO nanoparticles [33,34]. The physical properties of the NAPs (size, shape, surface area), as well as their functionalization with the physiologically active phytomolecules (triterpene saponins) contained in the Escin component, could be linked to these significant cytotoxic effects. Escin, the primary constituent of Aesculus hippocastanum seeds, has been documented in several studies to possess powerful anti-cancer properties against breast cancer cells [35,36].

The enhanced permeability and retention mechanism employed by these nanoparticles allows them to attach to receptor-binding ligands found on the tumor cell surface. MgO NPs can deliver chemotherapy drugs to the intended cancerous cell for effectiveness and reduced toxicity [37]. Moreover, the biocidal capability of MgO NPs can be enhanced upon encapsulation of a Chitosan substrate, as it also has effective anticancer activity [34]. 

MDA-MB-231 cell lines from breast cancer were treated with MgO-Chitosan-Pluronic F127-Escin NPs (1.56–200 µg) for 24, 48, and 72 h. The cells were subjected to MTT assay, and the values were depicted as ±SD of three individual experiments.

### 3.7. Evaluation of Apoptosis by AO/EB Dual Staining

ROS accumulation within the cells generally affects their membranes which leads to apoptosis-related changes in the cell membranes [38]. In MDA-MB-231 cells treated with MCsPFE NPs, apoptosis-associated membrane deformity was investigated by dual AO/EB labeling (Figure 6). To detect the apoptosis-related modifications in cell membranes throughout the apoptotic process, dual staining was performed. The AO dye can stain nuclear DNA that still has its whole cell membranes, while the EB dye can only stain cells that have lost their complete cell membrane [39]. The presence of AO-positive, yellowish, and green-colored cells following NP treatments demonstrated early apoptosis. Still, EB-positive cells were also found in the test groups, suggesting late apoptotic or dead cells upon treatment with MCsPFE NPs.

### 3.8. Analysis of Δψm by JC-1 Staining

It has been demonstrated that mitochondrial impairment contributes to apoptosis initiation and may be a key factor in the apoptotic pathway. According to reports, the Δψm loss occurs early in the apoptotic cell mechanism [40]. Examining Δψm to assess mitochondrial viability and functioning utilizes the JC-1 dye [41]. Cells containing high membrane potential exhibit intense green fluorescence, as observed in control cells (Figure 7). It was discovered that, after exposure to the MCsPFE NPs, Δψm was reduced, as demonstrated by the decreased fluorescence compared to control cells. Thus, the initiation of the apoptotic process could be confirmed after nanoparticle treatment of breast cancer cells.

### 3.9. Detection of ROS by DCFH–DA Staining

ROS may trigger cell death in a number of ways and are involved in both cell proliferation and apoptosis [42]. Different nanoparticles have been documented to trigger oxidative stress in vitro and in vivo [43,44]. Therefore, we examined the ROS-inducing capability of MCsPFE NPs by DCFH–DA staining to determine the cause of cytotoxicity observed in MDA-MB-231 cells. With a higher concentration of NPs, an increased green fluorescence was observed, indicating a higher ROS generation in the treated cells (Figure 8). When intracellular ROS builds up, it results in the loss of Δψm, leading to apoptosis. Through the production of ROS in human breast tumor cells, oxidative stress contributes to MCsPFE NP-induced cytotoxicity. Additionally, we noticed that MCsPFE NPs produced ROS in MDA-MB-231 cells, indicating that these NPs triggered oxidative stress-mediated cell death.

### 3.10. DAPI Staining

To distinguish apoptosis from necrosis, it is essential to assess it. One of the notable characteristics of cell death’s apoptotic mechanism is nuclear fragmentation. Therefore, the fluorescent DNA-binding agent DAPI was employed to examine apoptosis-related cell death and morphological modifications in cells [45]. The apoptotic ability of the NPs was demonstrated by the fragmented apoptotic bodies and shrunk nuclei in the treated cells, compared to the healthy and enlarged nucleus in the control cells (Figure 9). Thus, the findings concluded that the NPs augmented the ROS levels in the cells, disrupting the Δψm and leading to the initiation of the intrinsic apoptotic cell death pathway.

## 4. Conclusions

In summary, green MgO-Chitosan-Pluronic F127-Escin NPs have been synthesized and characterized using various techniques. Further, its antibacterial, antifungal, and anticancer activities have been examined. The average crystallite size of the green MCsPFE NPs was 46 nm, and an FCC crystalline structure was observed. From the FESEM and TEM images, the synthesized NPs exhibit a nanorod-like structure. Chemical compositions were identified by EDAX spectra. The MgO, Chitosan, Pluronic F-127, and Escin functional groups were employed by the FTIR spectrum. Various surface defects were identified by PL spectra. It was observed that the green MCsPFE NPS had high antimicrobial efficacy against yeast strains and Gram-positive and Gram-negative bacteria. Additionally, the NPs exhibited a significant cytotoxic effect against MDA-MB-231 cell lines, proven by MTT assay, dual AO/EB, JC-1, DCFH–DA, and DAPI staining procedures. The findings concluded that the NPs augmented the ROS levels in the cells and altered the Δψm, leading to the initiation of the intrinsic apoptotic cell death pathway. Thus, green MCsPFE NPs possess immense potential to be employed as an effective antimicrobial and anticancer treatment option. Future studies will evaluate the molecular-level mechanism of cell death in breast cancer cells in vivo.

## Figures and Tables

**Figure 1 nanomaterials-13-01227-f001:**
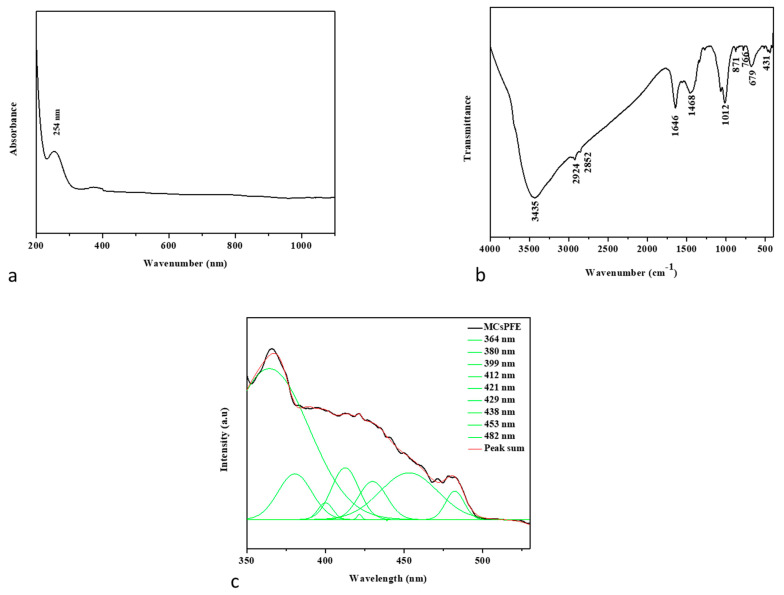
Spectral analysis of MgO-Chitosan-Pluronic F127-Escin NPs. UV-Vis spectrum of MgO-Chitosan-Pluronic F127-Escin NPs (**a**). FTIR Transmittance vs. wavenumber chart of MgO-Chitosan-Pluronic F127-Escin NPs derived from infrared analysis (**b**). Photoluminescence spectra for MgO-Chitosan-Pluronic F127-Escin NPs at room temperature (**c**).

**Figure 2 nanomaterials-13-01227-f002:**
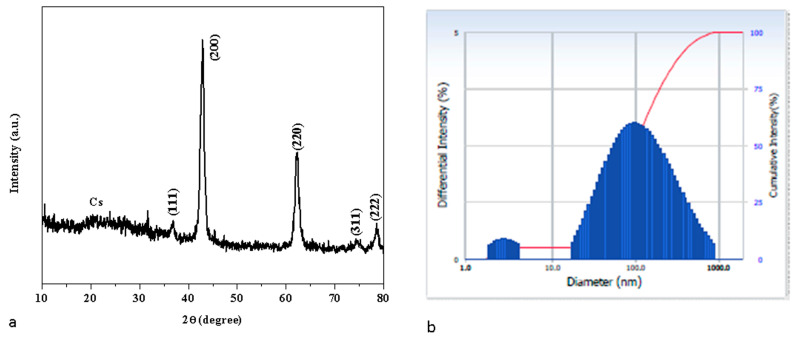
XRD Pattern of MgO-Chitosan-Pluronic F127-Escin NPs (**a**). Number-weighted particle size distribution obtained by DLS (**b**).

**Figure 3 nanomaterials-13-01227-f003:**
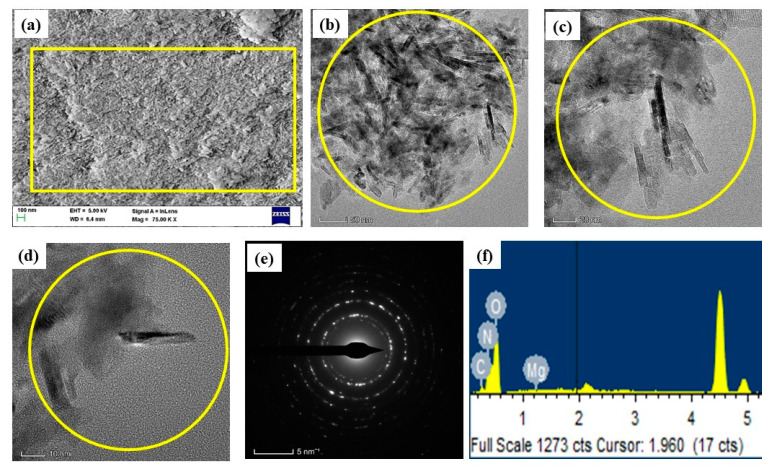
Electron Microscopic pattern of the MgO-Chitosan-Pluronic F127-Escin NPs: Lower and higher magnification FESEM image (**a**,**b**) TEM image (**c**,**d**), SAED pattern (**e**) and EDAX spectrum (**f**) of MgO-Chitosan-Pluronic F127-Escin NPs. Scale bar; (a) = 100 nm, (b) = 50 nm, (c) = 20 nm, (d) = 20 nm, (e) = 5 nm.

**Figure 4 nanomaterials-13-01227-f004:**
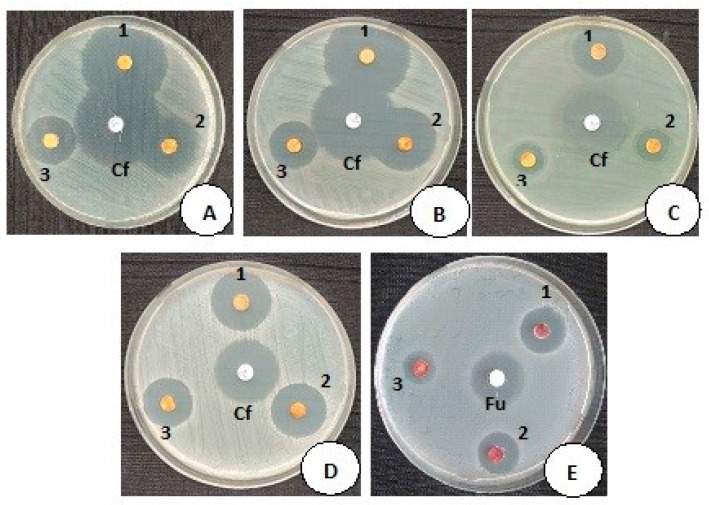
Antimicrobial activity of MgO-Chitosan-Pluronic F127-Escin NPs, activity against (**A**). *Escherichia coli* (ATCC: 25922); (**B**). *Enterobacter cloacae* (Clinical isolate); (**C**)*. Pseudomonas aeruginosa* (ATCC: 27853); (**D**)*. Staphylococcus aureus* (ATCC: 25923) and yeast fungi, (**E**). *Candida albicans* (ATCC: 10231). The MgO-Chitosan-Pluronic F127-Escin NPs concentration from 1–3 is successively diluted from 2 mg/mL; 1.5 mg/mL, and 1 mg/mL, respectively. (Cf and Fu are control antibiotic discs Ciprofloxacin Cf: 5 µg; and Fluconazole Fu: 25 µg).

**Figure 5 nanomaterials-13-01227-f005:**
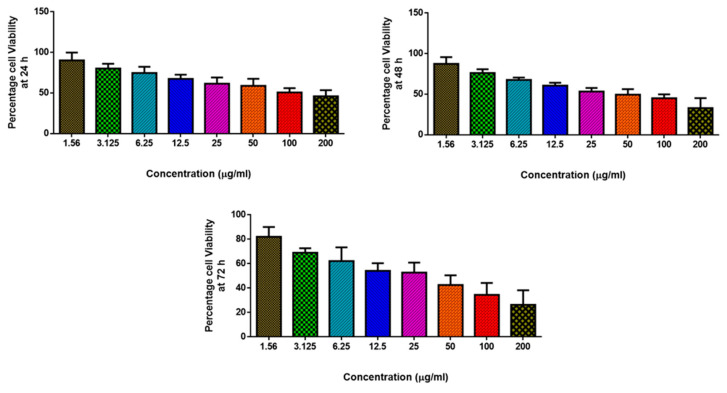
MgO-Chitosan-Pluronic F127-Escin NPs cause cytotoxicity in MDA-MB-231 cells.

**Figure 6 nanomaterials-13-01227-f006:**
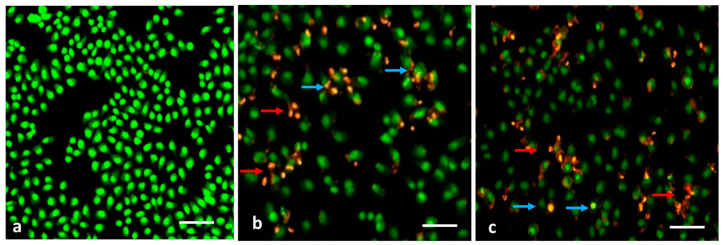
MDA-MB-231 cells exhibit apoptotic cell death induced by MgO-Chitosan-Pluronic F127-Escin NPs. MDA-MB-231 cell lines of breast cancer were treated with MgO-Chitosan-Pluronic F127-Escin NPs for 24 h. Acridine orange and ethidium bromide (1:1) was used to stain the cells, and representative stained cells from each group were shown. Control (**a**), 10 µg/mL of MgO-Chitosan-Pluronic F127-Escin NPs-treated cells (**b**) and 20 µg/mL of MgO-Chitosan-Pluronic F127-Escin NPs-treated cells (**c**). Green color cells indicate normal cells, yellowish green cells indicate early apoptosis (Blue arrow), yellow color cells indicate late apoptotic cell death (red arrow), and red cells indicate necrotic cell death (scale bar 50 μm). 20× magnification was used to capture the images.

**Figure 7 nanomaterials-13-01227-f007:**
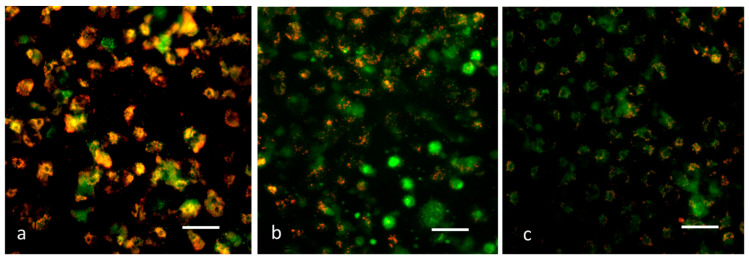
MDA-MB-231 cells with MgO-Chitosan-Pluronic F127-Escin NPs present have decreased mitochondrial membrane permeability. MgO-Chitosan-Pluronic F127-Escin NPs were used to treat breast cancer MDA-MB-231 cell lines. JC-1 was used to stain the cells, and a representative stained cell was shown for each group. Control (**a**), 10 µg/mL of MgO-Chitosan-Pluronic F127-Escin NPs-treated cells (**b**), and 20 µg/mL of MgO-Chitosan-Pluronic F127-Escin NPs-treated cells (**c**). Red fluorescence indicates normal mitochondria, and green fluorescence indicates damaged mitochondria (scale bar 50 μm). 20× magnification was used to capture the images.

**Figure 8 nanomaterials-13-01227-f008:**
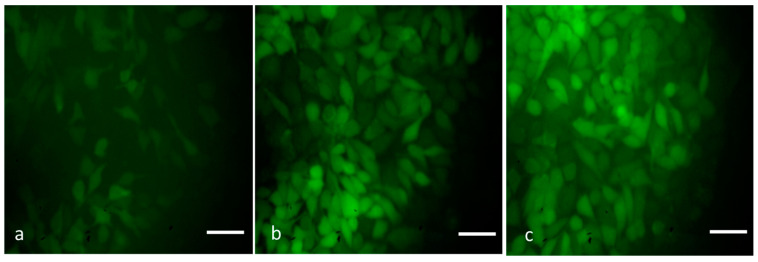
The MDA-MB-231 cell line is subjected to oxidative stress induced by MgO-Chitosan-Pluronic F127-Escin NPs. MgO-Chitosan-Pluronic F127-Escin NPs were used to treat breast cancer MDA-MB-231 cell lines. A representative stained cell for each group was shown against DCFH-DA stained cells. Control (**a**), 10 µg/mL of MgO-Chitosan-Pluronic F127-Escin NPs-treated cells (**b**), and 20 µg/mL of MgO-Chitosan-Pluronic F127-Escin NPs-treated cells (**c**). 20× magnification was used to capture the images (scale bar 50 μm).

**Figure 9 nanomaterials-13-01227-f009:**
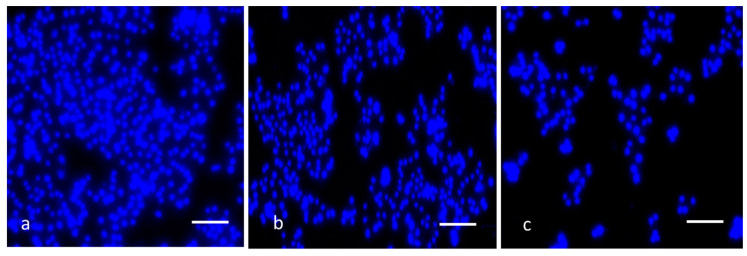
MDA-MB-231 cells with MgO-Chitosan-Pluronic F127-Escin NPs have altered nuclear integrity. MgO-Chitosan-Pluronic F127-Escin NPs were used to treat breast cancer MDA-MB-231 cell lines. DAPI was used to stain the cells, and a representative stained cell was shown for each group. Control (**a**), 10 µg/mL of MgO-Chitosan-Pluronic F127-Escin NPs-treated cells (**b**), and 20 µg/mL of MgO-Chitosan-Pluronic F127-Escin NPs-treated cells (**c**). 20× magnification was used to capture the images (scale bar 50 μm).

**Table 1 nanomaterials-13-01227-t001:** Quality control study of standard antibiotic discs by disc and antimicrobial activity of MgO-Chitosan-Pluronic F127-Escin NPs.

Reference ID	Microbial Strains	Control Antibiotic Disc Ciprofloxacin (5 µg) Zone of Inhibition (mm) *	CLSI Reference Range Expected Zone Based on Disc Potency *	2 mg/mL Zone of Inhibition (mm)	1.5 mg/mL Zone of Inhibition (mm)	1 mg/mL Zone of Inhibition (mm)
A	*E. coli* ATCC:25922	34.33 ± 1.41	30–40	34.33 ± 0.53	27.66 ± 0.53	22.66 ± 1.06
B	*Enterobacter cloacae* (Clinical isolate)	36 ± 0.92	-	37.33 ± 0.53	26.33 ± 0.53	21 ± 0.92
C	*P. aeruginosa* ATCC:27853	32.33 ± 0.53	25–33	24 ± 0.92	17.66 ± 0.53	15.66 ± 0.53
D	*S. aureus* ATCC: 25923	25.33 ± 0.53	22–30	23.66 ± 0.53	21.33 ± 0.53	19.33 ± 0.53
E	*Candida albicans* ATCC:10231 (Yeast fungi)	Control antifungal disc Fluconazole (25 µg) 26.33 ±0.53	25–30	19.33 ± 0.53	18.66 ± 1.06	16.33 ± 0.53

* The control antimicrobial discs Ciprofloxacin (5 µg) and Fluconazole (25 µg) zone of inhibitions were within the expected reference zones as per the guidelines of Clinical and Laboratory Standards Institute (CLSI).

## Data Availability

Not applicable.

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
