# Peer review of "Biopolymer Chitosan Surface Engineering with Magnesium Oxide-Pluronic-F127-Escin Nanoparticles on Human Breast Carcinoma Cell Line and Microbial Strains"

_nanomaterials, 2023, doi:10.3390/nano13071227_

Round 1

Reviewer 1 Report

In the present manuscript the authors report the synthesis and characterization of green MgO-Chitosan-Pluronic F127-Escin nanomaterial to be employed as antimicrobial and anticancer materials. The work is well performed and the biological results obtained are outstanding, therefore in my opinion the manuscript deserves to be published on Nanomaterial. However, some aspects should be clarified before publication.

- Several typos are present, please read carefully the manuscript and correct them

- The obtained nanomaterial was thoroughly investigated by several techniques, but the way how the characterization is represented is a bit confusing. I suggest the author to describe the results not in paragraph. Currently it seems a mere list of techniques. Furthermore, to better emphasize their findings, the characterization of pristine components should be added as well.

- Figure with better resolution should be provided. For example scale bar in FESEM and TEM images are barely distinguishable.

- In Figure 4, number of significant digit should be normalized.

Author Response

Answers to Reviewer’s Comments

The authors thank the reviewer and the editor for the valuable comments and suggestions of the manuscript, which have greatly improved the quality of the Manuscript. The following is the comment and the revisions made:

Reviewer: 1

  1. Several typos are present, please read carefully the manuscript and correct them.

Answer: The authors thank the reviewers for their valuable suggestions. As per the reviewer’s suggestion, we corrected all the typos throughout the manuscript. And the changes were marked in red text.

  1. The obtained nanomaterial was thoroughly investigated by several techniques, but the way how the characterization is represented is a bit confusing. I suggest the author to describe the results not in paragraph. Currently it seems a mere list of techniques. Furthermore, to better emphasize their findings, the characterization of pristine components should be added as well.

Answer: As per the reviewer's suggestion, the revised manuscript clearly explains the characterization.

  1. Figure with better resolution should be provided. For example scale bar in FESEM and TEM images are barely distinguishable.

Answer: As per the reviewer's suggestion, in the revised manuscript changed the Figure resolution.

  1. In Figure 4, number of significant digit should be normalized.

Answer: As per reviewer’s suggestion. We removed the digit from the fig.4 in the revised manuscript.

Reviewer 2 Report

The manuscript contains a large number of typographical errors, inaccuracies and poor quality of English.

For example, you need to correct the lower or upper apostrophe on the following lines:

100, 107, 111, 168, 205, 239 and so on

146-148 it is necessary to draw up the formula correctly

276-277 it is necessary to draw up the formula correctly

Low quality of all drawings in Fig .3.

There is no scale bar in all micrographs of cells.

Micrographs of cells with stained nuclei (DAPI staining) do not provide any reliable information about the development of apoptosis.

The MDA-MB-231 cells in Figure 6 look strange. Usually this culture has a developed micromorphology and spreading. On the presented micrographs (including those in the control), the cells look dumbfounded, which indicates their poor condition.

The authors did not provide an analysis of the accumulation of MnO nanoparticles in cells. There are no data on the efficiency of accumulation of MnO nanoparticles by cells of this type; therefore, it is impossible to speak with confidence about the cytotoxic and anticancer effect without studying the mechanisms of endocytosis and the efficiency of accumulation in cells.

The authors studied the level of intracellular ROS using DCF, but presented only microphotographs (Figure 8), which do not give an idea of the increase in the level of ROS after incubation. A positive control (such as hydrogen peroxide) should be given and fluorescence growth should be quantified. It is also worth noting that the authors used 10mM dye, which is incorrect and too much. These data look very strange - in the methods section it is written that the cells were removed from the plastic and stained, however, the presented results show spread cells (Figure 8), stained attached to the plastic.

For the analysis of apoptosis (Figure 6) the authors used a mixture of dyes acridine orange/ethidium bromide, which is not correct for the analysis of apoptosis. These dyes show the ratio of living and dead cells. For the analysis of apoptosis, it is necessary to use dyes for caspase 3, 7 or 8 or for example labeled annexin, which binds to phosphotylserine on the surface of apoptotic cells.

JC-1 dye was used to analyze the level of membrane mitochondrial potential (Figure 7). A positive control (eg, valinomycin or CCCP, FCCP) is required, as well as quantitation in the form of fluorescence intensity of JC-1 monomers or aggregates.

Author Response

Answers to Reviewer’s Comments

The authors thank the reviewer and the editor for the valuable comments and suggestions of the manuscript, which have greatly improved the quality of the Manuscript. The following is the comment and the revisions made:

Reviewer -2

  1. The manuscript contains a large number of typographical errors, inaccuracies and poor quality of English.

Answer: The authors thank the reviewers for their valuable suggestions. As per the reviewer’s suggestion, we corrected all the typos throughout the manuscript. And the changes were marked in red text.

  1. For example, you need to correct the lower or upper apostrophe on the following lines:

100, 107, 111, 168, 205, 239 and so on

Answer: As per the reviewer’s suggestion, we corrected all the chemical formulas, the numerals should be the subscript and the symbol throughout the manuscript. And the changes were marked in red text.

146-148 it is necessary to draw up the formula correctly

Answer: As per the reviewer's suggestion, the revised manuscript Tauc plot with the equation clearly rewritten

276-277 it is necessary to draw up the formula correctly

Answer: As per the reviewer's suggestion, the revised manuscript Tauc plot with the equation clearly rewritten

  1. Low quality of all drawings in Fig .3.

Answer: As per the reviewer's suggestion, in the revised manuscript changed the Figure resolution.

  1. There is no scale bar in all micrographs of cells.

Answer: Asper reviewer’s suggestion we included scale bar for all the micrographs.

  1. Micrographs of cells with stained nuclei (DAPI staining) do not provide any reliable information about the development of apoptosis.

Answer: The authors thank the reviewers for their valuable suggestions. The treated cells show the nuclear damage with increased fluorescence compare with control cells.

  1. The MDA-MB-231 cells in Figure 6 look strange. Usually this culture has a developed micromorphology and spreading. On the presented micrographs (including those in the control), the cells look dumbfounded, which indicates their poor condition.

Response: The authors thank the reviewers for their valuable suggestions. We obtained the MDA-MB-231 cells from the authenticated cell line repository and we cultured the cells. The MDA-MB-231 cells looks healthy and shows the clear morphology.

  1. The authors did not provide an analysis of the accumulation of MnO nanoparticles in cells. There are no data on the efficiency of accumulation of MnO nanoparticles by cells of this type; therefore, it is impossible to speak with confidence about the cytotoxic and anticancer effect without studying the mechanisms of endocytosis and the efficiency of accumulation in cells.

Answer: The authors thank the reviewers for their valuable suggestions. We obtained the MDA-MB-231 cells from the authenticated cell line repository and we cultured the cells. The MDA-MB-231 cells looks healthy and shows the clear morphology. In future we are planning for accumulation of nanoparticles in cells and apoptotic mechanism will be studied.

  1. The authors studied the level of intracellular ROS using DCF, but presented only microphotographs (Figure 8), which do not give an idea of the increase in the level of ROS after incubation. A Positive control (such as hydrogen peroxide) should be given and fluorescence growth should be quantified. It is also worth noting that the authors used 10mM dye, which is incorrect and too much. These data look very strange - in the methods section it is written that the cells were removed from the plastic and stained, however, the presented results show spread cells (Figure 8), stained attached to the plastic.

Answer: The authors thank the reviewers for their valuable suggestions. We obtained the MDA-MB-231 cells from the authenticated cell line repository and we cultured the cells. The MDA-MB-231 cells looks healthy and shows the clear morphology. In future studies authors will use positive control (such as hydrogen peroxide). As per reviewer’s suggestion 10mM dye, was a typo error. In revised manuscript we rectified the concentration and the changes were marked in red text.

Kim H, Xue X. Detection of Total Reactive Oxygen Species in Adherent Cells by 2',7'-Dichlorodihydrofluorescein Diacetate Staining. J Vis Exp. 2020 Jun 23;(160):10.3791/60682. doi: 10.3791/60682. PMID: 32658187; PMCID: PMC7712457.

  1. For the analysis of apoptosis (Figure 6) the authors used a mixture of dyes acridine orange/ethidium bromide, which is not correct for the analysis of apoptosis. These dyes show the ratio of living and dead cells. For the analysis of apoptosis, it is necessary to use dyes for caspase 3, 7 or 8 or for example labeled annexin, which binds to phosphotylserine on the surface of apoptotic cells.

The authors thank the reviewers for their valuable suggestions. We obtained the MDA-MB-231 cells from the authenticated cell line repository and we cultured the cells. The MDA-MB-231 cells looks healthy and shows the clear morphology. In future we are planning for study of caspase 3, 7 or 8, pro and anti-apoptotic proteins and apoptotic mechanism will be studied.

  1. JC-1 dye was used to analyze the level of membrane mitochondrial potential (Figure 7). A positive control (eg, valinomycin or CCCP, FCCP) is required, as well as quantitation in the form of fluorescence intensity of JC-1 monomers or aggregates.

The authors thank the reviewers for their valuable suggestions. We obtained the MDA-MB-231 cells from the authenticated cell line repository and we cultured the cells. The MDA-MB-231 cells looks healthy and shows the clear morphology.

Elefantova, K.; Lakatos, B.; Kubickova, J.; Sulova, Z.; Breier, A. Detection of the Mitochondrial Membrane Potential by the Cationic Dye JC-1 in L1210 Cells with Massive Overexpression of the Plasma Membrane ABCB1 Drug Transporter. Int. J. Mol. Sci. 2018, 19, 1985. https://doi.org/10.3390/ijms19071985.

Laha, D., Pal, K., Chowdhuri, A.R., Parida, P.K., Sahu, S.K., Jana, K., & Karmakar, P. (2019). Fabrication of curcumin-loaded folic acid-tagged metal organic framework for triple negative breast cancer therapy inin vitroandin vivosystems. New Journal of Chemistry.

Liao C, Xu D, Liu X, Fang Y, Yi J, Li X, Guo B. Iridium (III) complex-loaded liposomes as a drug delivery system for lung cancer through mitochondrial dysfunction. Int J Nanomedicine. 2018 Jul 30;13:4417-4431. doi: 10.2147/IJN.S170035. PMID: 30104875; PMCID: PMC6071621.

Jinfeng Zhang, Fang Fang, Bin Liu, Ji-Hua Tan, Wen-Cheng Chen, Zelin Zhu, Yi Yuan, Yingpeng Wan, Xiao Cui, Shengliang Li, Qing-Xiao Tong, Junfang Zhao, Xiang-Min Meng, and Chun-Sing Lee

ACS Applied Materials & Interfaces 2019 11 (44), 41051-41061

DOI: 10.1021/acsami.9b14552

Sivandzade, F., Bhalerao, A. and Cucullo, L. (2019). Analysis of the Mitochondrial Membrane Potential Using the Cationic JC-1 Dye as a Sensitive Fluorescent Probe. Bio-protocol 9(1): e3128. DOI: 10.21769/BioProtoc.3128.

Reviewer 3 Report

This article is devoted to modified magnesium oxide. The authors investigated the resulting product by various physicochemical methods, and also investigated its biological activity. The article presents a lot of experimental data and they are adequately described. As for the subject of the study, it is in trend, such works have been popular lately. However, there are some points for improvement:

1. Technical:

1.1 Increase the quality of your drawings.

1.2 If you are submitting microscopy data, indicate the magnification.

1.3 Unify the drawings. They must be in the same style.

1.4 In chemical formulas, the numerals should be the subscript and the symbol "-1" (for centimeters) the superscript.

2. Main:

2.1 Chitosan and its derivatives do not show up strongly in UV spectra. What was the point of doing this analysis?

2.2 X-ray diffraction. It is desirable to provide data for the original sample (chitosan).

2.3 Citation preferred: 10.1016/j.molstruc.2021.131083.

2.4 The main improvement that needs to be done is to increase the comparison with literature data for each method of analysis (both chemical and biological). This will help to adequately understand whether the authors have achieved what is already known (in terms of the properties and characteristics of the material) or not. In addition, it will allow authors and readers to understand further prospects and directions of research in this direction.

2.5 It is desirable to expand the conclusions.

Author Response

Answers to Reviewer’s Comments

The authors thank the reviewer and the editor for the valuable comments and suggestions of the manuscript, which have greatly improved the quality of the Manuscript. The following is the comment and the revisions made:

Reviewer -3

  1. Technical:

1.1 Increase the quality of your drawings.

Answer: As per reviewer’s suggestion we improved the quality of the drawings.

1.2 If you are submitting microscopy data, indicate the magnification.

Answer: As per reviewer’s suggestion we indicated the microscopy magnification and the changes were marked in red text.

1.3 Unify the drawings. They must be in the same style.

Answer: As per reviewer’s suggestion we arranged the drawings in same style.

1.4 In chemical formulas, the numerals should be the subscript and the symbol "-1" (for centimeters) the superscript.

Answer: As per the reviewer’s suggestion, we corrected all the chemical formulas, the numerals should be the subscript and the symbol throughout the manuscript. And the changes were marked in red text.

  1. Main:

2.1 Chitosan and its derivatives do not show up strongly in UV spectra. What was the point of doing this analysis?

Answer: Thamilarasan et al., reported chitosan nanocrystal UV-asborbition peaks at 250 nm. In the case of MCsPFE NPs chitosan subititution of MgO NPs absrobance peaks observed at 254 nm, due to the polymer subsititution effects this shift observed inthe MCsPFE NPs.

Thamilarasan, V., Sethuraman, V., Gopinath, K., Balalakshmi, C., Govindarajan, M., Mothana, R. A., ... & Benelli, G. (2018). Single step fabrication of chitosan nanocrystals using Penaeus semisulcatus: potential as new insecticides, antimicrobials and plant growth promoters. Journal of Cluster Science29, 375-384.

2.2 X-ray diffraction. It is desirable to provide data for the original sample (chitosan).

Chitosan peaks are matched with early literature; the same position of 2theta values is observed in the XRD results.

Karthikeyan, C., Varaprasad, K., Akbari-Fakhrabadi, A., Hameed, A. S. H., & Sadiku, R. (2020). Biomolecule chitosan, curcumin and ZnO-based antibacterial nanomaterial, via a one-pot process. Carbohydrate polymers249, 116825.

2.3 Citation preferred: 10.1016/j.molstruc.2021.131083.

Answer: As per the reviewer’s suggestion. We included the citation and marked in red text.

2.4 The main improvement that needs to be done is to increase the comparison with literature data for each method of analysis (both chemical and biological). This will help to adequately understand whether the authors have achieved what is already known (in terms of the properties and characteristics of the material) or not. In addition, it will allow authors and readers to understand further prospects and directions of research in this direction.

Answer: As per the reviewer's suggestion, in the revised manuscript, both chemical and biological properties match early literature.

2.5 It is desirable to expand the conclusions.

Answer: As per the reviewer's suggestion, in the revised manuscript conclusion are expand.

Round 2

Reviewer 2 Report

.

Author Response

Dear professor, 

Thank you very much for your valuable time and appreciable comments to enhance the quality of manuscript.  We have carried-out all your comments and academic editors comments. 

Many thanks again  

Reviewer 3 Report

Accepted

Author Response

Dear professor, 

Thank you very much for your valuable revision to enhance the quality of manuscript.  We have carried-out all your comments and academic editors comments. 

Many thanks again